# TRAIL Triggers CRAC-Dependent Calcium Influx and Apoptosis through the Recruitment of Autophagy Proteins to Death-Inducing Signaling Complex

**DOI:** 10.3390/cells11010057

**Published:** 2021-12-25

**Authors:** Kelly Airiau, Pierre Vacher, Olivier Micheau, Valerie Prouzet-Mauleon, Guido Kroemer, Mohammad Amin Moosavi, Mojgan Djavaheri-Mergny

**Affiliations:** 1Institut Bergonié, INSERM U1218, University of Bordeaux, 33000 Bordeaux, France; kelly.airiau@gmail.com (K.A.); pierre.vacher@inserm.fr (P.V.); valerie.prouzet-mauleon@u-bordeaux.fr (V.P.-M.); 2Lipides, Nutrition Cancer, INSERM, UMR1231, 21079 Dijon, France; Olivier.micheau@inserm.fr; 3UFR Science de Santé, Université de Bourgogne, Franche-Comté, 21079 Dijon, France; 4Centre de Recherche des Cordeliers, INSERM UMRS 1138, Sorbonne Université, Université de Paris, Equipe 11 Labellisée par la Ligue Contre le Cancer, 75006 Paris, France; kroemer@orange.fr; 5Metabolomics and Cell Biology Platforms, Institut Gustave Roussy, 94805 Villejuif, France; 6Pôle de Biologie, Hôpital Européen Georges Pompidou, AP-HP, 75015 Paris, France; 7Department of Molecular Medicine, National Institute of Genetic Engineering and Biotechnology, Tehran P.O. Box 14965/161, Iran

**Keywords:** ATRA, ATG7, autophagy, cancer, CRAC channels, DISC, leukemia, ORAI1, p62/SQSTM1, resistance to therapy

## Abstract

Tumor necrosis factor-related apoptosis-inducing ligand (TRAIL) selectively kills various cancer cell types, but also leads to the activation of signaling pathways that favor resistance to cell death. Here, we investigated the as yet unknown roles of calcium signaling and autophagy regulatory proteins during TRAIL-induced cell death in leukemia cells. Taking advantage of the Gene Expression Profiling Interactive Analysis (GEPIA) project, we first found that leukemia patients present a unique TRAIL receptor gene expression pattern that may reflect their resistance to TRAIL. The exposure of NB4 acute promyelocytic leukemia cells to TRAIL induces intracellular Ca^2+^ influx through a calcium release-activated channel (CRAC)-dependent mechanism, leading to an anti-apoptotic response. Mechanistically, we showed that upon TRAIL treatment, two autophagy proteins, ATG7 and p62/SQSTM1, are recruited to the death-inducing signaling complex (DISC) and are essential for TRAIL-induced Ca^2+^ influx and cell death. Importantly, the treatment of NB4 cells with all-*trans* retinoic acid (ATRA) led to the upregulation of p62/SQSTM1 and caspase-8 and, when added prior to TRAIL stimulation, significantly enhanced DISC formation and the apoptosis induced by TRAIL. In addition to uncovering new pleiotropic roles for autophagy proteins in controlling the calcium response and apoptosis triggered by TRAIL, our results point to novel therapeutic strategies for sensitizing leukemia cells to TRAIL.

## 1. Introduction

Tumor necrosis factor (TNF)-related apoptosis-inducing ligand (TRAIL, also known as TNFSF10/Apo2L) is a member of the TNF superfamily that can lead to the induction of cell death and non-cell death pathways [1,2,3]. The binding of TRAIL to its cognate receptors enables a plethora of cellular responses that can either contribute to tumor immunosurveillance or tumor development and metastasis in a cancer context-dependent manner [2,3,4]. TRAIL signaling is mainly transmitted through its interaction with receptor homotrimers: TRAIL-R1 (DR4) and TRAIL-R2 (DR5) contribute to extrinsic pathway apoptosis, while TRAIL-R3 (DcR1) and TRAIL-R4 (DcR2) act mostly as putative TRAIL-neutralizing decoy receptors [5,6]. Mechanistically, the binding of TRAIL to its cognate receptors DR4 or DR5 induces the assembly of a death-inducing signaling complex (DISC), in which the apoptosis initiator procaspase-8 (proCASP-8) is brought close of the intracellular death domain (DD) of DR4 or DR5 by the adaptor protein Fas-associated death domain (FADD), leading to the autocatalytic cleavage of CASP-8 and the subsequent activation of execute CASP-3, -6 or -7 [5,7]. Besides apoptosis induction, TRAIL can induce necroptosis [8] or autophagy, where it may function as either a promotor or inhibitor of cell death, depending on the cellular contexts [9,10,11]. Therefore, TRAIL can trigger several distinct cell death and non-cell death signaling pathways, which closely related to the recruitment of specific proteins and the assembly of DISCs [12,13].

Autophagy is an evolutionarily conserved catabolic process that eliminates cytoplasmic entities (e.g., macromolecules and organelles) to sustain cell survival under stressful conditions, including starvation, endoplasmic reticulum (ER) stress and chemotherapy. Mechanistically, the class III PI3K complex and BECN1 (ATG6) initiate phagophores formation as a platform for the engulfment of cytoplasmic components. Then, two ubiquitin-like pathways, the ATG5–ATG12 conjugate pathway and the microtubule-associated protein 1B light chain 3 (LC3B) conjugation system, participate in phagophore elongation by forming a closed double-membrane structure called an autophagosome [14]. In the ATG5–ATG12 conjugate pathway, ATG12 is activated by the ubiquitin-E1-like molecule ATG7 and then transferred to ATG10 to conjugate with ATG5 and constitute a large complex with ATG16. In the second system, LC3/ATG8 is cleaved by the protease ATG4 to form LC3B-I, after which it is activated by ATG7 and transferred to ATG3 to bond to phosphatidylethanolamine (PE) and form LC3B-II, which inserts into the autophagosome membranes. The recruitment of autophagic cargo is mediated by the autophagy receptor p62/SQSTM1 (p62), which interacts with the ubiquitinated proteins and LC3B [15]. However, beyond their conserved role in the autophagy process, autophagy proteins can also function in vesicular trafficking pathways and other signal transduction pathways, such as cell cycle and cell death [16,17]. In the context of cancer, autophagy acts as either a tumor promotor or a tumor suppressor, depending on the type and stage of the tumors [18]. In addition, the ambiguous contribution of autophagy to cytoprotective and lethal signaling pathways may be leveraged in cancer therapies [18,19,20].

In addition to their role in igniting lethal signaling, death receptors can transmit signals that favor cell survival and migration, for instance, by activating the PI3K/Akt and calcium (Ca^2+^) release signaling [6,21,22]. Calcium responses are sequentially initiated through phospholipase C γ1 (PLCγ1)-dependent inositol trisphosphate (IP3) production, IP3 receptor (IP3R) activation and the release of calcium ions (Ca^2+^) from the ER [23]. Subsequently, the decrease in the ER Ca^2+^ store is sensed by stromal interaction molecule 1 (STIM1), which, in turn, binds to the plasma membrane-located ORAI1 family proteins and stimulates store-operated Ca^2+^ entry, or capacitive Ca^2+^ influx, via the opening of Ca^2+^ release-activated Ca^2+^ (CRAC) channels [24]. Depending on the type of cells and stimuli, ryanodine receptors (RyR) can also operate as ER Ca^2+^ release channels [25].

The possibility of targeting TRAIL has attracted much attention due to the capacity of agonistic antibodies to selectively kill tumor cells [4,6]. However, its clinical application has been disappointing due to intrinsic or acquired resistance mechanisms [5,26]. In this study, we aimed to explore the role of calcium signaling in controlling the cell death induced by TRAIL in order to develop more potent TRAIL-based combination therapies. We selected the promyelocytic leukemia (APL)-driven NB4 cell line, which is highly responsive to differentiation therapy with all-*trans* retinoic acid (ATRA) [27] and which is relatively insensitive to TRAIL [28]. ATRA not only induces autophagy and differentiation in APL cells [29,30,31,32,33], but also tends to activate apoptosis, at least in part, through the upregulation of TRAIL [34,35]. Our results revealed that TRAIL triggers an initial Ca^2+^ efflux from ER stores, followed by a CRAC-dependent Ca^2+^ influx. In addition, we uncovered a new implication of ATG7 and p62 for regulating TRAIL-mediated calcium entry and apoptosis, which is mediated through their recruitment to DISCs. Interestingly, TRAIL sensitized NB4 cells to the apoptotic and autophagic responses induced by ATRA and *vice versa*, indicating that the combination of both drugs is particularly efficient.

## 2. Materials and Methods

### 2.1. Reagents

All-*trans* retinoic acid (#R2625), xestospongin C (#X2628), U733122 (#U6881), thapsigargin (#67526-95-8) and Sepharose 6B (#9012-36-6) were purchased from Sigma-Aldrich, Saint-Quentin-Fallavier, France. BTP2, N-[4-[3,5-Bis(trifluoromethyl)-1H-pyrazol-1-yl] phenyl]-4-methyl-1,2,3-thiadiazole-5-carboxamide (#3939) and 2-APB (2-aminoethoxydiphenyl borate) (#1224) were purchased from TOCRIS Bioscience, Noyal Châtillon sur Seiche, France. Tetra-methylrhodamine methyl ester (#T-668) and propidium iodide dye (#P4864) were purchased from Life Technologies, Saint-Aubin, France. Fura-2AM (#0230) was purchased from TEFLabs, Austin, TX, USA. Protein G-Sepharose beads were provided by Amersham Biosciences, Les Ullis, France. The antibodies anti-ORAI1 (#O8264) and anti-LC3 (#L7543) were from Sigma-Aldrich, Saint-Quentin-Fallavier, France; anti-STIM1 (#5668) and cleaved caspase-3 (Asp175) (#9661) were from Cell Signaling Technology, Danvers, MA, USA; anti-p62/SQSTM1 (#610832) and (#BML-PW9860) were from BD Biosciences, Le Pont de Claix, France and Enzo Life Sciences, Villeurbanne, France, respectively; anti-ATG7 (#sc-8668), anti-FADD (#sc-5559) and anti-caspase 8 (#sc-6136) were from Santa Cruz Biotechnology Inc., Santa Cruz, CA, USA; anti-caspase 8 (# MO32) was from MBL, MA, USA; anti-DR4 (#AB16955) and anti-DR5 (#AB16942) were from Chemicon, Millipore, Molsheim, France; horseradish peroxidase (HRP)-conjugated related secondary antibodies, including mouse IgG1 (#1070-05), mouse IgG2a (#1080-05), mouse IgG2b (#1090-05) and goat IgG (#6300-05), were purchased from Southern Biotech, Birmingham, AL, USA. Cell culture media were purchased from Life Technologies-Invitrogen, Villebon-sur-Yvette, France. Flag-tagged recombinant soluble human TRAIL and His-tagged human TRAIL were produced and used as described previously [36].

### 2.2. Expression Analysis of TRAIL Receptors in Cancers

To evaluate the expression pattern of TRAIL-Rs across cancers, we used the Gene Expression Profiling Interactive Analysis (GEPIA) web-based tool (http://gepia.cancer-pku.cn/index.html (accessed on 12 June 2021), which uses RNA sequencing expression data from The Cancer Genome Atlas (TCGA) for tumor and adjacent tumor samples, and data from the Genotype-Tissue Expression (GTEx) project for healthy normal tissues. The differential expression genes (DEGs) data for 28 cancers with at least 10 normal samples were downloaded for 28 or matched GTEx samples. These cancers included adrenocortical carcinoma (ACC), bladder urothelial carcinoma (BLCA), breast invasive carcinoma (BRCA), cervical squamous cell carcinoma and endocervical adenocarcinoma (CESC), colon adenocarcinoma (COAD), lymphoid neoplasm diffuse large B-cell lymphoma (DLBC), esophageal carcinoma (ESCA), glioblastoma multiform (GBM), head and neck squamous cell carcinoma (HNSC), kidney chromophobe (KICH), kidney renal clear cell carcinoma (KIRC), kidney renal papillary cell carcinoma (KIRP), acute myeloid leukemia (LAML), brain lower grade glioma (LGG), liver hepatocellular carcinoma (LIHC), lung adenocarcinoma (LUAD), lung squamous cell carcinoma (LUSC), ovarian serous cystadenocarcinoma (OV), pancreatic adenocarcinoma (PAAD), prostate adenocarcinoma (PRAD), rectum adenocarcinoma (READ), skin cutaneous melanoma (SKCM), stomach adenocarcinoma (STAD), testicular germ cell tumors (TGCT), thyroid carcinoma (THCA), thymoma (THYM), uterine corpus endometrial carcinoma (UCEC) and uterine Carcinosarcoma (UCS). The numbers and information related to the samples of this study have been summarized in the Appendix A. To generate a heatmap and the expression matrix, the complex heatmap R package version 2.7.1.1009 (https://jokergoo.github.io/ComplexHeatmap-reference/ accessed on 6 December 2020) based on the hierarchical clustering of genes and cancers was used. The q-value < 0.01 and log_2_FC > 1 (upregulated) or log_2_FC < 1 (downregulated) were considered to be significant values.

### 2.3. Cell Culture

The NB4 acute promyelocytic leukemia-derived cell line was gifted by Dr. M Lanotte [37]. The human leukemic T cell lymphoblast Jurkat cell line was purchased from the American Type Culture Collection. The cell lines used in this study were grown at 37 °C in a humidified atmosphere with 5% CO_2_ in RPMI 1640 culture media supplemented with 10% fetal bovine serum, 100 units/mL penicillin, 100 mg/mL streptomycin and 2 mM glutamine (Life Technologies-Invitrogen, Villebon-sur-Yvette, France). For this study, we used Jurkat T cells stably expressing either a dominant negative mutant of ORAI1 (Orai1E106A) or a small hairpin (sh) RNAmir-pGIPZ vector for ORAI1 (RHS4430-98715881 and -101067842) (Open Biosystems, Inc., Huntsville, AL, USA) that were previously generated and validated in the Patrick Legembre laboratory, as described in [38].

### 2.4. Generation of Autophagy Deficient Leukemia Cells

The pLenti CRISPR (pXPR) vectors expressing Cas9 and the single guide (sg)RNAs that specifically target ATG7 or non-specific sgRNA controls were constructed and validated as described in [39]. ATG7 knocked-out (KO) NB4 cells were generated using clustered regularly interspaced short palindromic repeats (CRISPR)/Cas9 system using these lentivirus vectors as we described in our previous work [40]. Briefly, NB4 cells were plated in a six-well plate (10^5^ cells/well) and after 24 h, they were infected with the expressing vectors (sgATG7 or sgcontrol). At 24 h post-transfection, they were cultivated in methylcellulose dishes and then selected with puromycin (2 μg/mL). Isolated colonies were then taken and seeded in a 96-well plate and cultivated for 2–3 weeks. The efficacy of the ATG7 depletion in NB4 cells was then determined using Western blotting.

For silencing p62/SQSTM1 protein expression levels, we used the small hairpin (sh)RNA lentivirus transduction approach as previously described by our group [31]. Briefly, two pLKO1 lentiviral vectors specifically expressing shRNA p62 #1 and shRNA p62 #2 as well as a scramble shRNA control (all from Open Biosystems, Huntsville, AL, USA) were used and the efficacy of p62 depletion in NB4 cells was then determined as described in [31].

### 2.5. Cell Death Assay

Cell death was determined by staining the cells either with propidium iodide (PI) dye for detecting the plasma membrane permeability or with tetramethyl rhodamine methyl ester perchlorate (TMRM) dye for evaluating the mitochondrial transmembrane potential (ΔΨm), as previously described [30]. Briefly, supernatant and attached cells were collected, pelleted at 350 g for 5 min and stained with either 1 μg/mL PI for 15 min or 150 nM TMRM for 30 min at 37 °C. Cells were then analyzed via flow cytometry using BD FACSCalibur (Becton Dickinson, Franklin Lakes, NJ, USA).

### 2.6. Measurement of Intracellular Calcium Concentration

The intracellular Ca^2+^ concentrations [Ca^2+^]i were ratiometrically measured via fluorescence video cell imaging using Fura-2AM, a leakage resistant calcium probe Fluo2-AM, as previously reported [38,41]. Briefly, NB4 or Jurkat cells were cultured on glass coverslips in six-well plates and then loaded with the 2 μM Fura-2AM for 30 min at 37 °C in Hank’s Balanced Salt Solution (HBSS) supplemented with 10 mM 4-(2-hydroxyethyl)-1-piperazineethanesulfonic acid (HEPES) buffer. After rinsing the cells and replacing the medium with HBSS, they were stimulated with TRAIL or Thapsigargin (TG) for the indicated period of times and placed under an invert fluorescence microscope (Olympus IX70). The images of the Fura-2AM fluorescence probe (λexc = 488 nm, λem = 515 nm) were obtained using a CoolSNAP fx CCD (charge coupled device) camera (digital camera, 12 bits), which records the fluorescence emitted by the cells in the form of gray levels, at the rate of one image every 10 s. The images were then analyzed for the quantification of fluorescence variations with software (Metafluor, Molecular Devices, San Jose, CA, USA). Between 20 and 30 cells were studied per each field (at least three fields at three different days were measured). An increase in the fluorescence emitted by this cell permeable probe (F/F0) reflects an increase in cytosolic Ca^2+^.

### 2.7. Western Blotting

The control and treated cells were incubated in the lysis buffer (Tris/SDS buffer, 10 mM Tris, pH 7.4, 1% SDS) supplemented with a cocktail of protease and phosphatase inhibitors and then treated with benzonase endonuclease (Sigma-Aldrich, Saint-Quentin-Fallavier France) for 5 min at room temperature. About 20–50 μg proteins of cell extracts were subjected to SDS-PAGE electrophoresis and then transferred to a Hybond-C super nitrocellulose membrane (#10600001, GE Healthcare, Life sciences, NY, USA), as previously described [31]. Protein bands were visualized using an ECL kit from GE Healthcare New York, NY, USA. Protein loading was checked through the staining of the membranes with Ponceau Red S (Sigma-Aldrich, Saint-Quentin-Fallavier, France). Membranes were then incubated with primary antibodies using the manufacturer’s instruction protocols followed by the addition of appropriate HRP-conjugated secondary antibody. The densitometry quantification was carried out using ImageJ software. All of the experiments were repeated at least two times and representative images have been presented.

### 2.8. Immunoprecipitation and DICS Assay

For DISC analysis, NB4 cells (10^8^ cells) were stimulated with 8 μg of His-TRAIL in 10 mL culture medium (800 ng/mL) for the indicated times at 37 °C. Cells were then washed with cold phosphate saline buffer (PBS) and lysed in 1 mL lysis buffer containing 1% NP-40, 20 mM Tris-HCl pH 7.5, 150 mM NaCl, 10% glycerol and a proteinase inhibitor cocktail. Cell lysates were then pre-cleared with Sepharose 6B for 1 h at 4 °C and after centrifugation, TRAIL receptor DISCs were immunoprecipitated overnight at 4 °C in the presence of protein G-Sepharose beads and an anti-caspase-8 targeting antibody. Finally, immunoprecipitated beads were washed four times with lysis buffer, eluted in a loading buffer (63 mM Tris-HCl pH 6.8, 2% SDS, 0.03% phenol red, 10% glycerol, 100 mM DTT), boiled for 5 min at 95 °C and processed for immunoblot analysis for the determination of the immunoprecipitated DISC components.

### 2.9. Statistical Analysis

Unless otherwise stated, the two-tailed ANOVA test was used for calculation of *p*-values for comparisons between two groups using GraphPad Prism 8 software. Results are the mean of three independent experiments ±S.D., each performed in triplicates. The *p*-values less than 0.05 were considered to be statistically significant.

## 3. Results and Discussion

### 3.1. Pan-Cancer RNA Sequencing Analysis Suggests TRAIL Resistance in Leukemia

We first evaluated the expression pattern of all four TRAIL-Rs across 28 of the cancer types from The Cancer Genome Atlas (TCGA), comparing them to their corresponding normal samples as mentioned in the Materials and Methods section (Figure 1). Leukemia, as exemplified by acute myeloid leukemia (abbreviated as LAML), has a unique TRAIL-R expression pattern that may reflect the resistance of these cells to TRAIL (Figure 1). Indeed, apoptotic TRAIL-R1 (DR4) was downregulated, while decoy receptors TRAIL-R3 and -R4 were significantly upregulated compared to normal blood cells (Figure 1). These results are fully in agreement with a previous report that showed that the downregulation of TRAIL-R1, along with the upregulation of TRAIL decoy receptors, are associated with resistance to TRAIL-mediated apoptosis in leukemia cells [28].

### 3.2. TRAIL Triggers CRAC-Dependent Ca^2+^ Influx in NB4 and Jurkat Leukemia Cells: A Mechanism That Protects Cells against Apoptosis

Recent reports revealed that TRAIL-R1 (DR4) and TRAIL-R2 (DR5) have distinct roles in the control of apoptosis and calcium signaling [25,41]. We sought to further analyze the connection between TRAIL and Ca^2+^ signaling in two leukemia cell lines, Jurkat and NB4 cells, and thus investigated the possible implication of STIM1/ORAI1-coupled CRAC channels, as well as the inositol 1,4,5-trisphosphate 3 receptor (IP3R), a well-known Ca^2+^ release channel at the endo/sarcoplasmic reticulum. Following the treatment of NB4 cells with TRAIL, an immediate rise in the intracellular concentration of Ca^2+^ [Ca^2+^]i followed by a sustained plateau was recorded (Figure 2A,B). Jurkat T cell leukemia cells, as a well-known model of calcium signaling, also demonstrated a similar trend, with a rise in Ca^2+^ upon TRAIL treatment (Figure 2C), suggesting that an early increase in cytosolic Ca^2+^ is generally activated upon TRAIL exposure. To decipher the role of IP3R, we preincubated NB4 cells with either U73122, a PLCγ1 inhibitor, or xestospongin C (XesC), a specific inhibitor of IP3R that blocks the rapid ER-dependent increase in [Ca^2+^]i. Both inhibitors totally abrogated [Ca^2+^]i augmentation (Figure 2A). In addition, NB4 cells pretreated with N-[4-[3,5-Bis(trifluoromethyl)-1H-pyrazol-1-yl] phenyl]-4-methyl-1,2,3-thiadiazole-5-carboxamide (BTP2), an inhibitor of store-operated Ca^2+^ influx [42], prevented the cytoplasmic Ca^2+^ influx induced by TRAIL (Figure 2B). Similarly, Jurkat T cells depleted from PLCγ1 or ORAI1 failed to exhibit TRAIL-mediated ER calcium release and the subsequent sustained Ca^2+^ influx, respectively (Figure 2C). These results support the idea that upon TRAIL treatment, inositol 1,4,5-trisphosphate 3 (IP3) is generated by activated PLCγ1, and this leads to a rapid transient ER Ca^2+^ release into the cytosol via IP3R and the subsequent activation of STIM1/ORAI1-coupled CRAC channels that sustains the Ca^2+^ signal.

Depending on the context, the increase in [Ca^2+^]i exerts either pro-apoptotic or anti-apoptotic roles [43,44]. For example, it was shown that CD95- or TRAIL-mediated Ca^2+^ release led to apoptosis resistance due to delayed DISC formation [38] or TRAIL–DR endocytosis, respectively [25]. To determine if there is a connection between TRAIL-induced Ca^2+^ response and apoptosis in NB4 cells, we evaluated the effect of different Ca^2+^ blockers on TRAIL-induced apoptosis in NB4 cells. TRAIL used at 250 ng/mL was able to induce 30–33% apoptotic cell death in NB4 cells (Figure 2D,E). Interestingly, the ER calcium blockers of XesC and U73122 did not affect, or marginally dampened, TRAIL-induced apoptosis (Figure 2D), suggesting that IP3R-dependent ER Ca^2+^ release is uncoupled from apoptosis under this condition. On the contrary, we found that the pretreatment of cells with the ORAI1 channel inhibitor BTP2 significantly enhanced TRAIL-driven cell death, suggesting a contribution of CRAC-dependent Ca^2+^ influx to resistance to apoptosis induced by TRAIL in NB4 (Figure 2E) and Jurkat (Figure 2F) cells. Similarly, pretreatment with the extracellular Ca^2+^ chelator EGTA sensitized NB4 cells to TRAIL-induced apoptosis, supporting the idea that Ca^2+^ entry from extracellular sources plays an anti-apoptotic role in response to TRAIL (Appendix A). Such an anti-apoptotic role of CRAC channels has been reported in response to CD95L [38] and other conditions [45,46] in cancer cells. It is possible that the inhibition of capacitive Ca^2+^ influx by BTP2 results in a relative Ca^2+^ deficit in ER, which may lead to persistent ER stress, thereby provoking the induction of apoptosis [47].

### 3.3. ATG7 and p62 Are Both Required for Calcium Influx Induced by TRAIL

A recent report points to the central role of autophagy in controlling T cell receptor-mediated Ca^2+^ influx [48]. The relationship between Ca^2+^ signaling and autophagy is reciprocal; while autophagy modulation (both its inhibition and activation) by Ca^2+^ signaling has been well documented, the role of autophagy in regulating [Ca^2+^]i is poorly understood [49]. To understand whether the TRAIL-induced Ca^2+^ response and autophagy are interconnected, we used CRISPR/Cas9 and shRNA approaches to silence ATG7 and p62 expression levels, respectively (Figure 3A,B). At least three clones for sgATG7 (clones #1, #2 and #3) and two sources of NB4 cells that express two different shRNAs against p62 were used. As shown in Figure 3C,D, the removal of ATG7 and p62 significantly prevented a TRAIL-induced calcium response, comparable to the effects observed for U73122 and XesC (Figure 2A), implying that ATG7 and p62 are involved in the ER Ca^2+^ release into the cytosol and the subsequent CRAC-mediated Ca^2+^ influx. Similarly, the [Ca^2+^]i augmentation induced by TRAIL was dramatically impaired in the Jurkat cells that express an shRNA against p62 compared to control cells (Appendix A), suggesting that p62-dependent calcium release is a general response elicited by TRAIL.

**Figure 2 cells-11-00057-f002:**
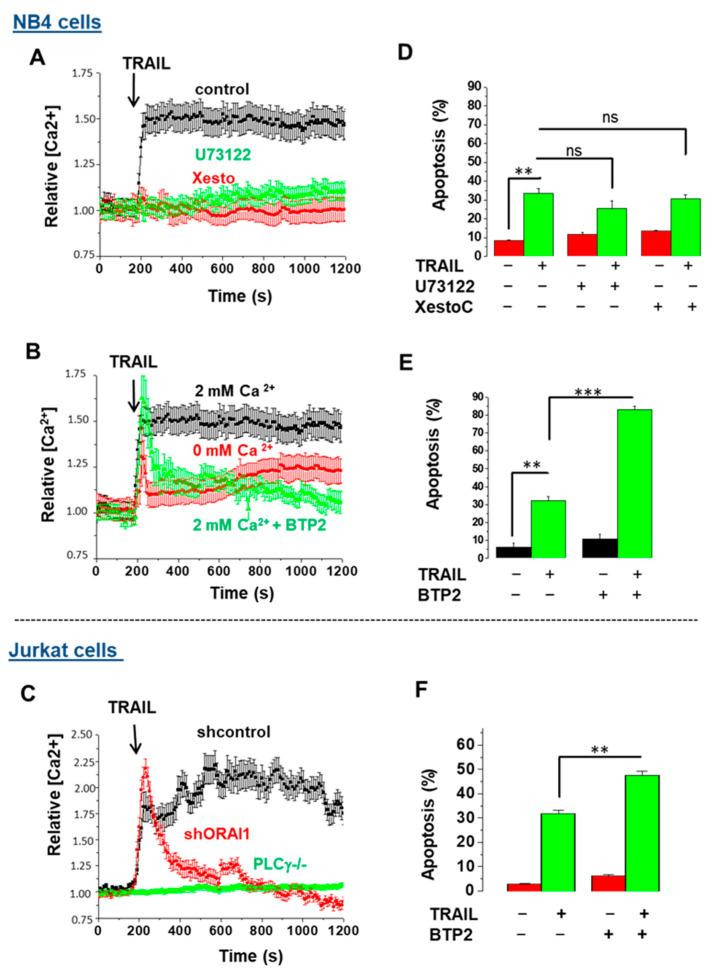
TRAIL-mediated Ca^2+^ influx prevents the induction of apoptosis in NB4 and Jurkat cells. NB4 cells were first treated with either 10 μM PLCγ1 inhibitor U73122, 2 µM specific inhibitor of IP3R xestospongin C (XesC) or 5 µM BTP2, an inhibitor of CRAC channel-mediated calcium entry 15 min prior treatment with TRAIL (250 ng/mL). (**A**,**B**) The calcium responses were measured after loading cells with Fura-2AM dye. (**D**,**E**) Apoptosis was evaluated via flow cytometry using TMRM staining after 24 h of treatment. (**C**) Jurkat control cells and PLCγ1/Orai1-depleted Jurkat T cells were treated with TRAIL (250 ng/mL) and then subjected to Ca^2+^ measurement as described in the Materials and Methods section. (**F**) Jurkat cells were first treated with 5 µM BTP2 and then TRAIL (250 ng/mL). Cell death was assessed through the measurement of the mitochondrial transmembrane potential using TMRM dye after 24 h. The percentage of cells exhibiting low TMRM staining were scored for each treatment. Results are from three independent experiments and expressed as means ± S.D. Non-significant (ns); statistically significant changes represented as ** *p* ≤ 0.001 and *** *p* ≤ 0.0001.

**Figure 3 cells-11-00057-f003:**
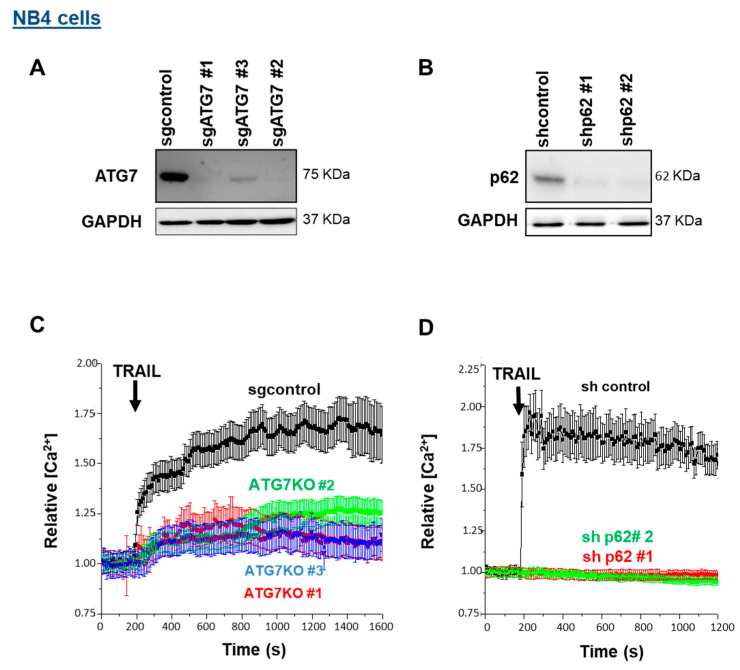
Effects of ATG7 and p62 depletion on TRAIL-induced Ca^2+^ influx in NB4 cells. NB4 cells were depleted for ATG7 (**A**) and p62 (**B**) as mentioned in the Materials and Methods section. ATG7 knocked-out (KO) NB4 cells were generated by CRISPR/Cas9 system using sgATG7 #1–3 and sgcontrol lentivirus vectors. For p62/SQSTM silencing, two lentiviral vectors specifically expressing shRNA p62 #1 and shRNA p62 #2 were used. (**C**,**D**) ATG7-depleted NB4 cells and NB4 cells expressing shp62 were exposed to 250 ng/mL TRAIL and relative [Ca^2+^]i levels were evaluated via fluorescence cell imaging after the Fura-2AM-staining of cells.

### 3.4. ATG7 and p62 Are Recruited to DISC to Regulate TRAIL-Induced Apoptosis

Next, we questioned whether ATG7 and p62 modulate the initial events of TRAIL-induced apoptosis, and thus investigated the composition of DISCs followed caspase-8 immunoprecipitation in the NB4 cells treated with TRAIL. In line with earlier reports [5,7], we found that TRAIL promotes the recruitment of the well-known components of DISCs, including CASP-8, FADD, DR4 and DR5, in NB4 cells. Interestingly, we observed that ATG7 and p62 were also recruited to DISCs upon TRAIL treatment (Figure 4A).

We then investigated in more detail the role of ATG7 and p62 in the induction of apoptosis by using ATG7-depleted NB4 cells and NB4 cells that express shRNA against p62. Interestingly, the depletion of both ATG7 and p62 led to a noticeable inhibition of TRAIL-induced apoptosis, suggesting that these autophagy proteins not only control Ca^2+^ influx, but also contribute to TRAIL-induced apoptosis (Figure 4B,C). Because ATG7 and p62 silencing was unable to enhance TRAIL-driven cell death (Figure 4B,C) unlike the CRAC channel inhibitor of BTP2 (Figure 2E,F), we hypothesize that this response is uncoupled from their role in the regulation of Ca^2+^ influx induced by TRAIL. These data collaborate with the study showing the pro-death effect of ATG7 in response to HW1, a human single-chain variable fragment (scFv) that specifically binds to TRAIL-R2 [50].

One mechanism that could explain the regulation of apoptosis by ATG7 and p62 could be the localization of ATG7 and p62 to the DISC which may facilitate the TRAIL-driven apoptotic signal. In the same line, ATG5, another autophagy protein, was shown to localize into the DISC and regulate components of the extrinsic apoptosis pathway via interaction with the death domain of FADD [51,52,53]. Accordingly, we also observed the recruitment of ATG5 to the DISC in NB4 cells treated with TRAIL (data not shown). Furthermore, our results may be interpreted in the context of an intracellular death-inducing signaling complex (iDISC), where the autophagosomal membrane serves as a platform for interaction between FADD and p62 with ATG5 and LC3-II, respectively, to mediate the recruitment and subsequent activation of proCASP-8 [54,55,56]. Accordingly, it was reported that the depletion of ATG7 prevents iDISC formation and subsequently CASP-8-dependent apoptosis in CHMP2A-deficient osteosarcoma and neuroblastoma cells [55]. Moreover, p62 aggregation was shown to promote the cell death induced by the BH3 mimetic agent ABT-263, as a consequence of CASP-8 aggregation and full activation [56,57]. Besides apoptosis, it was shown that the autophagic proteins can also control necroptosis [58] or influence other cancer-related pathways, such as migration and invasion, as reported in [59,60].

### 3.5. Autophagy Disruption Results in a Defect in ER Ca^2+^ Calcium Homeostasis

To gain more mechanistic insight into how autophagy regulates the TRAIL-induced Ca^2+^ response, we studied the effects of autophagy (ATG7) depletion on ER Ca^2+^ homeostasis. Recently, Jia et al. [48] reported that the impairment of the Ca^2+^ influx in autophagy-deficient Jurkat cells resulted from abnormal ER expansion and enhanced ER Ca^2+^ control and ATG7-KO NB4 cells with the Sarco/endoplasmic reticulum Ca^2+^ ATPase (SERCA) pump inhibitor thapsigargin (TG) [48]. TG depletes ER Ca^2+^ storage by blocking the ER calcium uptake mediated by the SERCA pump [61]. We used a low concentration of TG (5 nM) that could efficiently block the transports of Ca^2+^ from the cytosol into the ER, as evidenced by the subsequent increase in cytosolic Ca^2+^ levels (Figure 5A), but was unable to significantly stimulate ER stress-dependent apoptosis (Figure 5B) [61]. Notably, this response was more pronounced in ATG7-depleted NB4 cells, suggesting that more calcium content is stored in the ER of ATG7-depleted cells than that of control cells (Figure 5A). This enhanced Ca^2+^ content could be related to an abnormal ER expansion due to a defect of autophagy, as previously reported for the ATG7-deficient T lymphocytes activated via the T cell receptor [48].

We next investigated whether ATG7 loss can influence cell death in the NB4 cells treated by TG. While TG was unable to stimulate apoptosis in control NB4 cells, a significant induction of cell death was observed in ATG7-KO cell lines (Figure 5B). Thus, an increase in ER Ca^2+^ content resulting from ATG7 loss is associated with enhanced cell death in response to TG. It is possible that the induction of a prolonged unresolved ER stress in ATG7-depleted cells is responsible for the activation of cell death by TG as reported in [62]. Additional mechanisms have been proposed for the induction of cell death by TG which rely on DR5 upregulation, the non-autophagic function of LC3B and the involvement of unfolded protein response (UPR) mediators [63].

Thus, we hypothesize that ATG7 regulates TRAIL-mediated Ca^2+^ responses and cell death through distinct ways: (i) the regulation of ER organelle homeostasis and ER Ca^2+^ storage, and consequently Ca^2+^ influx into cells; (ii) favoring the recruitment of apoptotic components to DISCs, which may facilitate cell death induction; (iii) the distinct dysregulation of DR4 and DR5, which may consequently favor a calcium response and apoptotic DISC formation, respectively [41]. However, the precise mechanisms that couple ER Ca^2+^ release and apoptosis to autophagy and DISCs need further investigation.

### 3.6. ATRA Sensitizes NB4 Cells to TRAIL by Enhancing DISC Formation

Finally, we evaluated the sensitivity to TRAIL in APL cells that, according to our own results (Figure 1) and other reports [64], are invariably resistant to TRAIL-mediated apoptosis. It has been suggested that combination therapies may improve the efficacy of TRAIL in leukemia. For example, synergic effects were reported for the combination of ATRA with arsenic trioxide (ATO) or TRAIL in leukemic cells [34,65,66]. As shown in Figure 6A, time course experiments revealed that NB4 cells were relatively resistant to TRAIL-induced apoptosis even up to 48 h of treatment. Western blot analysis confirmed a low activation of caspases, especially CASP-3 (as revealed by the presence of cleaved forms of caspase 3), after 48 h (Figure 6B), further confirming that TRAIL is unable to cause an efficient level of apoptosis in NB4 cells.

Many reports demonstrated that FasL/CD95 and TRAIL induce both autophagy and apoptosis [10,67], where autophagy can either promote or inhibit apoptosis [17,68]. The selective autophagy-mediated degradation of a negative or positive regulator of apoptosis, which differently affect the receptor for FasL and TRAIL, may explain this dual role of autophagy [17,58,68]. In addition, we and others have previously shown that ATRA promotes autophagy induction and the accumulation of p62 in APL cells [31,32]. Here, we found that both TRAIL and ATRA elicited a significant accumulation of two well-known autophagy markers, LC3-II and p62, after 24 h and 48 h treatment of NB4 cells (Figure 6B). Because ATRA promotes the upregulation of CASP-8 and p62 as shown in [31,69] and in Figure 6B, we speculated that the combination of TRAIL and ATRA might facilitate apoptosis induction by enhancing the recruitment of p62 and CASP-8 into the DISC. As shown in Figure 6C, the addition of ATRA prior to TRAIL treatment markedly enhanced TRAIL-mediated CASP-8 processing and the assembly of DISC components (i.e., CASP-8, DR4, DR5 and FADD) in NB4 cells, which is in line with the massive induction of apoptosis observed under this condition (Figure 6A). Similarly, it has been reported that the combination of TRAIL and ATO promoted CASP8-dependent DISC formation and apoptosis in leukemia cells [65]. It should be mentioned that ATRA can also stimulate SOCE-dependent calcium influx in NB4 cells [70], and we observed that ORAI1 is recruited to the DISC when ATRA was added to TRAIL (data not shown). These results suggest a molecular connection between ORAI1 and DISC assembly. Whether or not this response is involved in the stimulatory effect of ATRA on TRAIL-driven apoptosis and DISC formation remains to be elucidated.

Taken together, we observed that ATG7 and p62 proteins are recruited to the DISC to form a large macromolecular complex that contributes to Ca^2+^ signaling responses and the apoptosis induced by TRAIL (Figure 2 and Figure 4). It is worth noting that a significant activation of autophagy (as evidenced by the accumulation of LC3B-II) is also detected after a relatively long (48 h) exposure to TRAIL (Figure 6). The accumulation of both autophagosomes and p62/SQSMT1 protein aggregates, but not autolysosomal activity, has been shown to be required for TRAIL-induced CASP-8 activation and apoptosis in TRAIL-sensitive cells [56,71]. Thus, the extent of autophagy levels in cells may determine CASP-8 activity and the sensibility of cells to TRAIL treatment [58]. Further investigation is needed to fully understand the reason for the cell type-specific differential response to TRAIL.

## 4. Conclusions

Altogether, we demonstrated herein a new role for the autophagy proteins ATG7 and p62 in regulating CRAC-dependent Ca^2+^ influx and ER calcium homeostasis in APL cells treated with TRAIL. We found that upon TRAIL treatment, ATG7 and p62 are recruited to the DISC and are involved in the induction of Ca^2+^ influx and apoptosis. Interestingly, the combination of TRAIL and ATRA led to a significant enhancement of DISC assembly and sensitized APL cells to TRAIL-induced cell death. Taken together, our results unveil a complicated multi-layer role for autophagic proteins in regulating Ca^2+^ signaling and the apoptosis triggered by TRAIL alone or in combination with ATRA. These findings may lead to new therapeutic strategies for combating the resistance of leukemic cells against TRAIL and ATRA.

## Figures and Tables

**Figure 1 cells-11-00057-f001:**
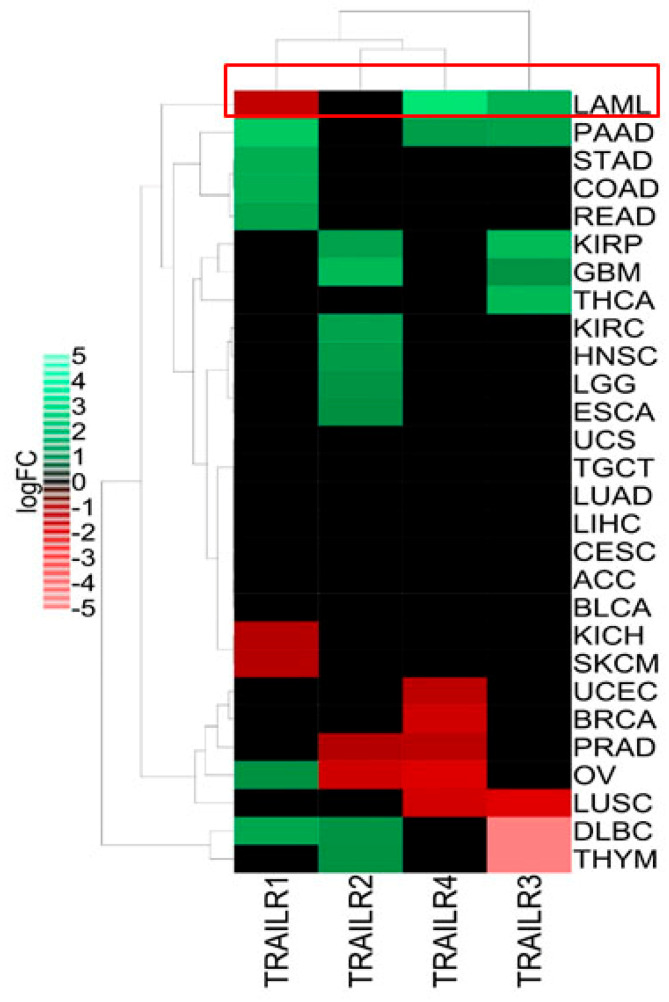
TRAIL receptors expression pattern across 28 TCGA cancer types. The RNA sequencing data for 28 TCGA (The Cancer Genome Atlas) cancers as well as normal adjacent tumors and normal GTEx (Genotype-Tissue Expression) samples were extracted from Gene Expression Profiling Interactive Analysis 2 (GEPIA2). Then, the differential mRNA expression levels (log_2_ fold change (FC)) of four well-known TRAIL-R1 (DR4), TRAIL-R2 (DR5), TRAIL-R3 (DcR1) and TRAIL-R4 (DcR2) were analyzed in each cancer compared to their corresponding normal samples. The heatmap construction and clustering were performed as mentioned in Materials and Methods. Green and red colors represent significant (*p*-value < 0.01) upregulation (log_2_FC > 1) or downregulation (log_2_FC < 1) of TRAIL-Rs in different cancers, respectively. TCGA cancer groups include BLCA (bladder urothelial carcinoma), BRCA (breast invasive carcinoma), CESC (cervical squamous cell carcinoma and endocervical adenocarcinoma), CHOL (cholangiocarcinoma), ESCA (esophageal carcinoma), GBM (glioblastoma multiforme), HNSC (head and neck squamous cell carcinoma), KIRC (kidney renal clear cell carcinoma), KICH (kidney chromophobe), KIRP (kidney renal papillary cell carcinoma), LAML (acute myeloid leukemia), LIHC (liver hepatocellular carcinoma), LUAD (lung adenocarcinoma), LUSC (lung squamous cell carcinoma), PAAD (pancreatic adenocarcinoma), PCPG (pheochromocytoma and paraganglioma), PRAD (prostate adenocarcinoma), SARC (sarcoma), SKCM (skin cutaneous melanoma), STAD (stomach adenocarcinoma), THCA (thyroid carcinoma), THYM (thymoma), UCEC (uterine corpus endometrial carcinoma). Leukemia (LAML) has been highlighted in a red rectangle.

**Figure 4 cells-11-00057-f004:**
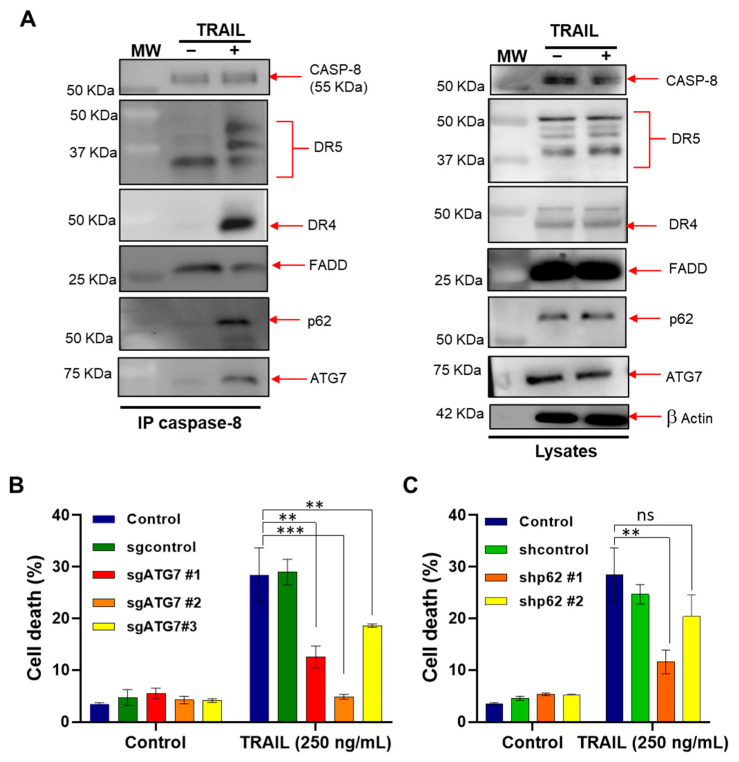
ATG7 and p62 both contribute to TRAIL-induced cell death. (**A**) NB4 cells were treated with His-TRAIL (800 ng/mL) for the indicated times and then subjected to cell lysis. Left panel, the DISCs were immunoprecipitated using an anti-caspase 8 antibody and then analyzed via Western blotting for the expression levels of the autophagy proteins (ATG7 and p62) as well as DISC components (CASP-8, DR4, DR5 and FADD). Right panel, total lysates were loaded as a control. A representative immunoprecipitation experiment is shown. MW: molecular weight of proteins. (**B**,**C**) ATG7 KO cells (**B**) and p62 KD (**C**) cells were treated with 250 ng/mL of TRAIL for 24 h and then apoptosis was evaluated via flow cytometry using TMRM dye. The results are from three independent experiments ± SD. Non-significant (ns); statistically significant changes represented as ** *p* ≤ 0.001 and *** *p* ≤ 0.0001.

**Figure 5 cells-11-00057-f005:**
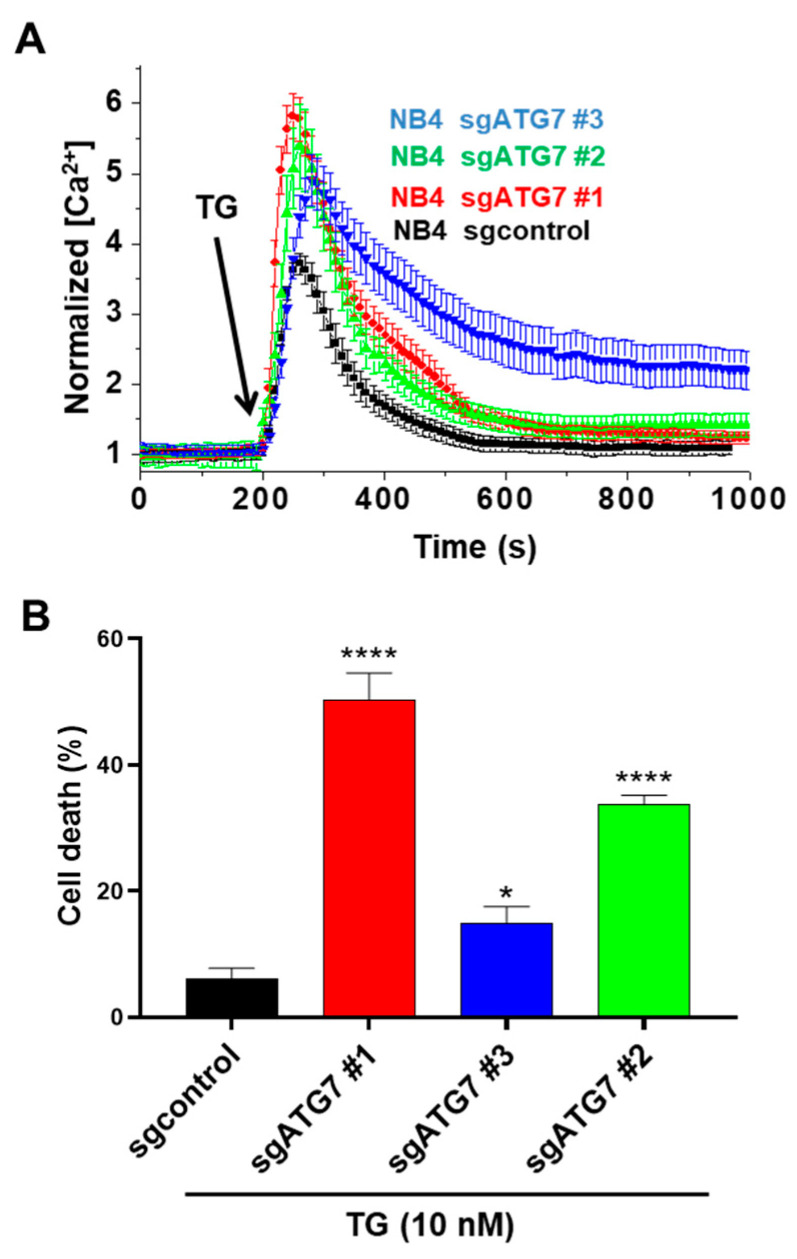
The effects of ATG7 disruption on calcium and apoptosis responses, alone or in the presence of sarco/endoplasmic reticulum Ca^2+^ ATPase pump inhibitor thapsigargin. (**A**) Control NB4 cells and ATG7-KO NB4 cells were incubated in a calcium-free medium and then stimulated with 5 nM (TG). Intracellular calcium levels were evaluated as described in the Materials and Methods section. Apoptosis was evaluated in control NB4 cells and the ATG7-KO NB4 cells treated with 10 nM TG for 24 h. In experiment (**B**), the results of three different ATG7 KO clones (#c1, #c2 and #c3) have been presented. All results are from three independent experiments (each performed in triplicates) and expressed as means ± SD. * and **** represent *p* ≤ 0.01 and *p* ≤ 0.0001, respectively.

**Figure 6 cells-11-00057-f006:**
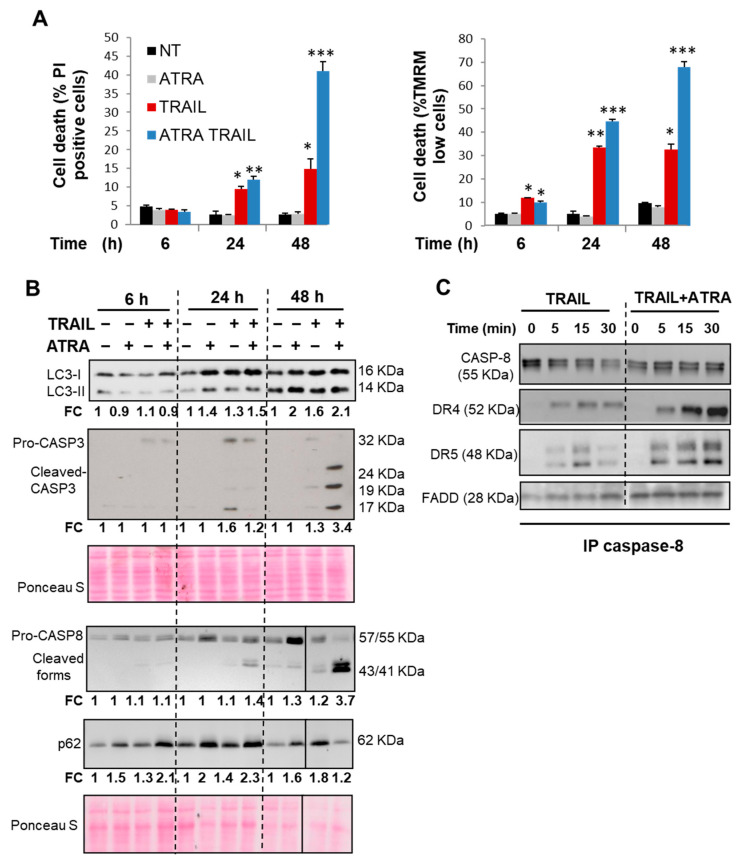
Effects of TRAIL, alone or in combination with ATRA on autophagy, DISC formation and cell death. NB4 cells were pretreated with 10 nM ATRA for 24 h alone or with TRAIL (250 ng/mL) for the indicated times (6–48 h). (**A**) Early apoptosis (MOMP) and late apoptosis (plasma membrane permeability) were then measured by staining the cells with TMRM and PI staining, respectively. The percentage of cells exhibiting low TMRM staining and high PI staining were scored for each condition. Results were from three independent experiments and expressed as means ± S.D. Statistically, changes in the treated groups (ATRA +/- TRAIL) compared with none-treated group (NT) have been represented as * *p* ≤ 0.01, ** *p* ≤ 0.001 and *** *p* ≤ 0.0001. (**B**) Immunoblot analyses of CASP-8 and CASP-3 processing and autophagy markers levels (LC3-II and p62) were shown. For the densitometry quantification, CASP-3 cleaved fragments, CASP-8 cleaved forms, p62 and LC3B-II/LC3B-I protein levels were normalized to corresponding Ponceau red S using the ImageJ software. The protein fold changes (FC) represent the ratio of treated cells versus control non-treated cells at each time point. (**C**) NB4 cells were pretreated with ATRA (10 nM) for 24 h and then treated with His-TRAIL (800 ng/mL) for the indicated time. After cell lysis, the DISCs were immunoprecipitated using an antibody against caspase 8 and analyzed by Western blotting for the expression levels of DISC components (CASP-8, DR4, DR5 and FADD).

## Data Availability

Data will be available on request.

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
