# Peer review of "TRAIL Triggers CRAC-Dependent Calcium Influx and Apoptosis through the Recruitment of Autophagy Proteins to Death-Inducing Signaling Complex"

_cells, 2021, doi:10.3390/cells11010057_

Round 1
Reviewer 1 Report
The study entitled, “TRAIL triggers CRAC-dependent calcium influx and apoptosis through the recruitment of autophagy proteins to death-induced signaling complex” is a highly novel and interesting approach to elucidating lesser known regulators of calcium signaling and how this can affect apoptosis and cell death in cancer. While the manuscript definitely provides a high overall merit and potential for publication, the present manuscript requires several changes before recommendation for publication:
Major:
Figure 2B: Better experiment would be to add BTP2 first. It is well-known that BTP2 will inhibit calcium entry, but if you add that first, then you can see if TRAIL is rescuing the effect of calcium entry inhibition. Adding BTP2 second is just blocking entry after calcium has already entered. If the goal is to sequester all of the calcium out of the cytosol after TRAIL addition, another experiment option would be to replace BTP2 with a chelator like EGTA. Currently, it is unclear how adding BTP2 second contributes new information to this figure.
Figure 2C: The reason why TRAIL induces increases in cytosolic calcium in the absence of Orai1 is unclear. Is this a store-independent process? The authors are trying to argue for a store-dependent process, and further attempt to argue this by manipulating PLCy. The explanation of the results in Figure 2C is inadequate since there is no clear explanation of the interpretation of why TRAIL leads to a cytosolic increase in calcium whereas PLCy knockout eliminates this effect. Further explanation or experimentation is recommended to clarify this point.
3.1 Results. Unclear, it states “LAML has a unique low TRAIL-R expression pattern that suggests TRAIL resistance”, yet it is unclear how Figure 1 shows a low “TRAIL-R” expression pattern. This sentence should either be taken out or there should be data to show that LAML has a low expression relative to other cancers/cell lines, small grammatical error “are associated with resistant (should be resistance) to TRAIL-mediated apoptosis in leukemia cells”
Minor:
3.2 Results. Grammatical error “DR4 and DR5 have distinct role (should be roles) in the control of…”
Figure 4: There are no indicated times on the figure contrary to the figure legend stating it.
Western blot validation: Why do all the western blots use ponceau S for loading control validation as opposed to more common single proteins like B-Actin or GAPDH?
Reviewer 2 Report
Airiau et al. investigated that TRAIL influences a calcium released-activated channel (CRAC)-dependent calcium influx relative to apoptosis and autophagy. You showed mainly calcium influx change with the Fura-2AM and presented the major data in Figures 2, 3, and 5. However, it is somewhat difficult to understand because the result to be proved is not systematic or robust.
So I would like to offer some comments.
Major
- In lines 28-29, the proposed purpose is very unclear. I wish I could re-write it.
- In lines 29-40, The results for the purpose should be summarized in detail.
- The introduction part is too long. Compared to the results to be proved, the introduction part is long, so it seems a bit difficult. Therefore, it would be better to reduce it to one page. Specifically, it is necessary to summarize and organize lines 65-89 and lines 90-118.
- In lines 104-106, I don't know why this part is necessary for The introduction part. I understand that TRAIL is clinically significant, but it is important because there has been no study related to calcium influx. Therefore, I think it is better to emphasize that point more.
- In lines 248-257, a clear description is required. This shows that DR4 and DR5 are very important in leukemia using several cancer cells, but the expression is rather difficult. It would be better to substitute Figure 1 with clear results.
- In NB4 leukemia cells, TRAIL concentrations of 8 μg and 250 ng/mL were used. Why was the concentration different?
- Figure 4 and 3.4. ATG7 and p62 are recruited to DISC to regulate TRAIL-induced apoptosis
In the results, the DISC proof experiment is quite different from Bellail et al. and Humphreys et al. Bellail AC, Tse MC, Song JH, Phuphanich S, Olson JJ, Sun SY, Hao C. DR5-mediated DISC controls caspase-8 cleavage and initiation of apoptosis in human glioblastomas. J Cell Mol Med. 2010;14:1303-17; Humphreys LM, Fox JP, Higgins CA, Majkut J, Sessler T, McLaughlin K, McCann C, Roberts JZ, Crawford NT, McDade SS, Scott CJ, Harrison T, Longley DB. A revised model of TRAIL-R2 DISC assembly explains how FLIP(L) can inhibit or promote apoptosis. EMBO Rep. 2020;21:e49254), so why was it performed in an unproven method? In particular, the results shown in the lysate are somewhat incomprehensible.
There was no change between the TRAIL-treated group and the non-treated group. If TRAIL is processed, shouldn't there be a difference in caspase-8? Other data seems to be the same. To increase the lysate, GAPDH or beta-actin as an internal control should be added.
- Figure 6 and 3.6. ATRA sensitizes NB4 cells to TRAIL by enhancing DISC formation
It isn't easy to understand why the ATRA addition test result is necessary. It just seems to have been added as an additive because there are few necessary results. More accurate experimental data for Figures 2, 3, 4 and 5 will be needed for accurate results demonstration.
In Figure 6, the data on the left side of B is under the same conditions, but was the order of the treatment groups shown in the wrong order? Showing the data also seems a bit beginner level. Internal control GAPDH or beta-actin should be added.
Minor
I would like to correct some grammatical errors.
Line 45, superfamily
Line 47, of cellular responses
Line 47, mainly transmitted
Line 50, contribute to extrinsic
Line 54, is brought close of the intracellular
Line 57, TRAIL can induce
Line 62, closely related to
Line 74, double-membrane structure called
Line 91, migration, for instance, by
Line 94, Calcium responses are
Line 106, help develop
Line 193, the efficacy of p62 depletion in NB4 cells was
Line 259, that protects cells
Line 330, response, comparable to the effects
Line 334, p62 compared to
Line 366, with the study
Line 381, Besides apoptosis,
Reviewer 3 Report
In the article entitled “TRAIL triggers CRAC-dependent calcium influx and apoptosis through the recruitment of autophagy proteins to death-inducing signaling complex", Airiau et al. analyzed the ability of TRAIL to induce CRAC-dependent calcium influx and apoptosis. This is an interesting study. Appropriate methodology has been employed and the conclusions appear to be justified based on the data at hand.
In order to better appreciate the data, some revisions are needed:
Major points:
- The authors use Ponceau Red S to check protein loading, but it would be better to use Actin or Tubulin as loading control. Please substitute the membranes stained with Ponceau Red S with membranes hybridized with an anti-Tubulin or anti-Actin antibody.
- In Figure legend 4, the authors indicate that cells were treated with 8 μg of TRAIL. What concentration does it correspond to?
- Figure 6, panel B: the treatments corresponding to 48 hours have not the same layout in the upper and lower panels. This may confuse the readers. I suggest changing the order in one of the two panels or alternatively label the lower panel with the letter C and rename panel C as panel D.
Minor points:
- Please correct typing errors through all the text and edit the English.
- Please add the missing cities in Materials and Method paragraph.
Reviewer 4 Report
In the article entitled “TRAIL triggers CRAC-dependent calcium influx and apoptosis through the recruitment of autophagy proteins to death-inducing signalling complex” by Airiau K. et al, the authors investigated the role of autophagy regulatory proteins ATG7 and p62/SQSTM1 in controlling cell death and calcium signalling induced by TRAIL, demonstrating a role for these two proteins in TRAIL-induced cellular effects. They also evaluated the combination of TRAIL and ATRA as a novel strategy to avoid drugs’ resistance.
To the reviewer’s knowledge, the topic of the manuscript is innovative; the subjects is attractive. The manuscript is well written.
However, in the reviewer’s opinion, there are some issues that should be addressed.
Major issue:
- In Figures 3A and 6B, the authors showed the Ponceau stained membranes as loading controls for Western Blot analysis. The reviewer’s opinion is that beta-actin should be used as loading control to normalize the levels of proteins for the proper interpretation of their levels’ modulation. Moreover, for all Western Blot data, densitometric analysis would make data clearer and their interpretation easier (the authors cited it in the “Materials and Methods” section, but it is not reported in figures or text).
Minor points:
- In the “Materials and Methods” section:
- please add the manufacturers data where they are missing;
- in paragraph 2.3, the author should avoid repeating all the cancers name that are already listed in Table S1;
- in paragraph 2.7, specify the Western Blot detection reagent used, if don’t indicate elsewhere.
- Figure 3 legend is not exhaustive: please describe more clearly the Figure.
- Figure 3D is not perfectly sharp and its quality needs to be enhanced.
- Along the text, there are some English mistakes and typing errors, please provide corrections.
Reviewer 5 Report
In this manuscript, Airiau et al, identified ATG7 and p62 as critical players recruited to the death-inducing signling complex, in response to TRAIL and leading to increased Ca2+ influx and consequent cell death.
This mechanism is interesting and overall well demontrated by data. However the authors claims (lines 368-369 and 462-463) that ATG7 and p62 proteins are recuited to the DISC to form a large macromolecular complex. I would be ineresting to show by immunofluorescence that in cells treated with TRAIL, all the components of the proposed molecular mechnism actually colocalize.
Round 2
Reviewer 2 Report
I think you should check again for grammatical errors.
Reviewer 3 Report
The article was improved substantially. Overall, in my opinion the revised version of the manuscript can be accepted for publication without additional modifications.
Reviewer 4 Report
The authors answered and argue properly to the reviewer's requests